# First Evidence of Gastroprotection by *Schinus molle*: Roles of Nitric Oxide, Prostaglandins, and Sulfhydryls Groups in Its Mechanism of Action

**DOI:** 10.3390/molecules27217321

**Published:** 2022-10-28

**Authors:** María Elena Sánchez-Mendoza, Yaraset López-Lorenzo, Leticia Cruz-Antonio, Daniel Arrieta-Baez, Miranda Carolina Pérez-González, Jesús Arrieta

**Affiliations:** 1Escuela Superior de Medicina, Instituto Politécnico Nacional, Plan de San Luis y Díaz Mirón, Colonia Casco de Santo Tomás, Miguel Hidalgo, Mexico City 11340, Mexico; 2Ingeniería en Tecnología Ambiental, Universidad Politécnica de Chiapas, Campus Suchiapa, Carretera Tuxtla Gutiérrez—Portillo Zaragoza km 21+500, Colonia Las Brisas, Suchiapa 29150, Mexico; 3Facultad de Estudios Superiores Zaragoza, UNAM, Av. Guelatao No. 66, Colonia Ejército de Oriente, Iztapalapa, Mexico City 09230, Mexico; 4Centro de Nanociencias y Micro y Nanotecnologías, Instituto Politécnico Nacional, Unidad Profesional Adolfo López Mateos, Av. Luis Enrique Erro s/n, Mexico City 07738, Mexico

**Keywords:** *Schinus molle*, traditional medicine, gastroprotection, gastric ulcers, bioassay-guided study, triterpene

## Abstract

*Schinus molle* is a plant traditionally used in Mexico to treat gastric disorders. However, no scientific evidence has been reported on its gastroprotective effect. The aim of the current contribution was to conduct a bioassay-guided study on *S. molle* to evaluate its gastroprotective activity in a model of Wistar rats given ethanol orally to induce gastric lesions. The hexane and dichloromethane extracts from the tested plant showed over 99% gastroprotection at a dose of 100 mg/kg. From the hexane extract, two of the three fractions (F1 and F2) afforded over 99% gastroprotection. The F1 fraction was subjected to column chromatography, which revealed a white solid. Based on the ESI-MS analysis, the two main compounds in this solid were identified. The predominant compound was probably a triterpene. This mixture of compounds furnished about 67% gastroprotection at a dose of 100 mg/kg. Pretreatment with L-NAME, indomethacin, and NEM was carried out to explore the possible involvement of nitric oxide, prostaglandins, and/or sulfhydryl groups, respectively, in the gastroprotective activity of the white solid. We found evidence for the participation of all three factors. No antisecretory activity was detected (tested by pylorus ligation). In conclusion, evidence is herein provided for the first time of the gastroprotective effect of *S. molle*.

## 1. Introduction

A peptic ulcer stems from a circumstantial loss of the epithelial mucosa in an area of the digestive tract exposed to gastric juice. A gastric ulcer is specific to the stomach. The lesion extends through the *muscularis mucosae* and reaches the submucosa, in some cases forming perforations that lead to peritonitis [1]. Gastric ulcers affect 5–10% of the world’s population at some point in life [2]. They have multiple etiologies and are associated with an imbalance between factors leading to and protective of the gastric mucosa [3]. Among the aggressive factors are physical stress, *Helicobactor pylory* infection, and the consumption of tobacco, alcohol, and certain types of medications, particularly nonsteroidal anti-inflammatory drugs [4]. The main protective factors are the bicarbonate mucus layer, the endogenous antioxidant system, prostaglandins, nitric oxide (NO), and blood flow [5].

The majority of drugs prescribed to treat this condition are expensive, and all have side effects when taken for a long period of time. They include antacids, cytoprotective agents, prostaglandin analogs, histamine H_2_-receptor antagonists, and proton-pump inhibitors [6]. Although the latter inhibitors (e.g., omeprazole) are the most promising for treating peptic ulcers, recent studies have shown that their prolonged use can cause irreversible adverse effects. For instance, by decreasing the absorption of vitamin B_12_, proton-pump inhibitors can engender dementia, neurological damage, anemia, hypergastrinemia [7], acute myocardial infarction [8], and pancreatic cancer [9].

The foregoing highlights the need to search for new therapeutic alternatives. Among the most important sources of new drugs are medicinal plants [10], which are the basis of primary health services for about 80% of the world population. As one example, *Schinus molle* is utilized in Mexican traditional medicine to treat infections of the urinary tract and of the respiratory, digestive, and genitourinary systems, and to treat skin ulcers (as a diuretic and tonic agent) and gastroduodenal disorders [11].

To our knowledge, no scientific report has yet validated or invalidated the empirical use of this plant to treat gastroduodenal disorders. Thus, the aim of the current contribution was to conduct a bioassay-guided study on *S. molle* to test the gastroprotective activity of its extracts and certain fractions in a model of Wistar rats administered ethanol to induce gastric lesions. An analysis was performed of the possible roles of NO, prostaglandins, and sulfhydryl groups in this gastroprotective effect. Anti-secretory activity was also evaluated by pylorus ligation.

## 2. Results

### 2.1. Gastroprotective Activity of the Hexane and Dichloromethane Extracts

Oral administration of the hexane and dichloromethane extracts of *S. molle* seeds at doses of 30 and 100 mg/kg inhibited ethanol-induced ulceration (Table 1). The maximum protective effect for each extract was observed at the 100 mg/kg dose (99.03 ± 0.28% and 99.32 ± 0.28% of gastroprotection, respectively). No significant difference existed between the effects of these two extracts at the 30 mg/kg dose or 100 mg/kg dose (Table 1). Moreover, no significant difference was detected between the effect of either extract (at 100 mg/kg) and that of carbenoxolone (the reference drug, at 100 mg/kg).

### 2.2. Gastroprotective Activity of the Fractions of the Hexane Extract

Three fractions (F1, F2, and F3) were obtained from the fractionation of the hexane extract by column chromatography, each of which was examined at a 100 mg/kg dose. Only F1 and F2 were active, and both reached almost 100% gastroprotection (Table 2). The percentage of gastroprotection of each of these compounds was the same as that of carbenoxolone, the reference drug (Table 2). Hence, F1 and F2 are apparently as efficacious as carbenoxolone. F3 was considered inactive, having provided less than 50% gastroprotection (Table 2).

### 2.3. Gastroprotective Activity of the White Solid

From F1, fractions 227–256 were obtained as a white solid, which was a mixture of compounds. The oral administration of the mixture significantly inhibited ethanol-induced lesions (Figure 1A). There was a maximum effect of 67.20 ± 5.64% protection at the dose of 100 mg/kg. However, compared to the 91.32 ± 1.10% gastroprotection obtained with 100 mg/kg of carbenoxolone, the reference compound proved to be more effective than the white solid (Figure 1B). Considering the gastroprotection of 64.71 ± 5.64% and 52.89 ± 6.09% with the doses of 30 and 10 mg/kg of the white solid, respectively, the effect was not significantly different between doses of this mixture of compounds, and thus it was not dose-dependent. There was no statistical difference between the gastroprotection afforded by the white solid and carbenoxolone at the doses of 10 and 30 mg/kg.

### 2.4. Mass Spectrometric Analysis of the White Solid

The white solid was analyzed by electrospray ionization (ESI) mass spectrometry (Figure 2), identifying the two principal compounds based on their molecular weight. The predominant one was probably a triterpene (compound I), and the other an aliphatic chain with ester and hydroxyl functional groups (compound II). The former compound (58.8% relative intensity) was a molecular ion at *m*/*z* 477.3319 [M + Na]^+^ (Calc. *m*/*z* 477.3339), which was correlated with a small peak at *m*/*z* 455.3521 that corresponds to [M + H]^+^ (Calc. *m*/*z* 455.3519). Both peaks are consistent with the molecular formula C_30_H_46_O_3_ (58.8%, Figure 3, compound I). The second major peak (22.75% relative intensity) was observed at *m*/*z* 381.2971 [M + Na]^+^ (Calc. *m*/*z* 381.2975), which is consistent with the molecular formula C_21_H_42_O_4_ (22.75%, Figure 3, compound II).

### 2.5. Participation of L-NAME, Indomethacin, and NEM in the Gastroprotective Effect of the White Solid

Given that N^G^-nitro-L-arginine methyl ester hydrochloride (L-NAME) is a nonspecific inhibitor of nitric oxide synthase (NOS), it was used to examine the participation of NO in the effect of the white solid (fractions 227–256 of F1). First, one group of rats was treated with 70 mg/kg of L-NAME plus the vehicle and another with the vehicle only. There was no significant difference between the two ulcer indices (128.64 ± 13.13 and 123.41 ± 11.88 mm^2^, respectively). Pretreatment with L-NAME inhibited the gastroprotective activity of the white solid (Figure 4A), indicating the involvement of NO in its mechanism of action [12]. The present results for L-NAME pretreatment followed by carbenoxolone treatment agree with reports in the literature [13].

To explore the possible contribution of prostaglandins to the mechanism of action of the white solid, animals were pretreated with indomethacin (a COX inhibitor). First, gastroprotection was evaluated after administering indomethacin plus the vehicle in one group of rats and the vehicle only in another group. There was no significant difference between the two ulcer indices (93.70 ± 6.85 and 91.94 ± 11.23 mm^2^, respectively). The ulcer index of the animals pretreated with indomethacin and then treated with the white solid (98.99 ± 7.84 mm^2^) was not significantly different from that of the vehicle control group (Figure 4B). Therefore, prostaglandins play a role in the mechanism of action of fractions 227–256 of F1. The gastroprotective effect of carbenoxolone was also attenuated by pretreatment with indomethacin (93.08 ± 7.84 mm^2^), in agreement with previously published data [13].

Animals were pretreated with *N*-ethylmaleimide (NEM), a blocker of sulfhydryl groups, to assess the participation of the latter in the gastroprotective activity of the white solid. First, one group of rats was treated with NEM plus the vehicle and another with the vehicle only. There was no significant difference between the two ulcer indices (126.78 ± 13.82 mm^2^ and 109.99 ± 8.66 mm^2^, respectively; Figure 4C). There was also a lack of any significant difference in the ulcer indices when comparing the animals receiving the NEM pretreatment followed by treatment with the white solid (98.91 ± 10.14 mm^2^) with those given the vehicle only. Hence, sulfhydryl groups are probably involved in the mechanism of action of fractions 227–256 of F1. Regarding carbenoxolone, the current results are in agreement with previous reports [13].

### 2.6. Antisecretory Activity

To measure the antisecretory activity of the treatments (Table 3), the volume and pH of the gastric content were measured. No difference in either parameter was detected between the control group and the animals administered the white solid. However, the comparison of the pH between the latter mixture of compounds and omeprazole showed a significant difference (Table 3). According to the results, the white solid does not have antisecretory activity.

## 3. Discussion

Peptic ulcers are characterized by inflamed lesions or excavations of the mucosa in the gastrointestinal system, and gastric ulcers are specific to the stomach. They are both caused by an imbalance between factors leading to and protective of the gastric mucosa [14]. The drugs presently available to treat gastric ulcers not only fall short of providing gastroprotection but have harmful side effects [15]. There is clearly a need to search for new drugs capable of eliciting better protection with minimal adverse events.

Medicinal plants constitute one of the main sources of new drugs. Although *S. molle* is traditionally used in Mexico to treat gastric disorders, no scientific study has yet confirmed or refuted its therapeutic benefit [11]. In the current contribution, the separate oral administration of the hexane and dichloromethane extracts of *S. molle* each prevented gastric damage when ethanol was subsequently given orally to the animals. Both extracts had the same effect at the two doses employed, providing over 99% gastroprotection at the highest dose (100 mg/kg; Table 1). Thus, the plant has more than one active compound.

Since the hexane extract was afforded in greater yield, it was subjected to fractionation by column chromatography. Of the three fractions obtained, F1 and F2 (but not F3) showed substantial activity, reaching almost 100% gastroprotection at 100 mg/kg (Table 2). A white solid, representing fractions 227–256, was isolated from F1 of the hexane extract. It also provided gastroprotection (~67%) at the 100 mg/kg dose. The ESI-MS spectrum of this white solid displayed two main peaks, attributed to molecular ions of *m*/*z* 477.3319 [M + Na]^+^ and *m*/*z* 381.2971 [M + Na]^+^. The former is consistent with the molecular formula of C_30_H_46_O_3_ (compound I). This probably corresponds to a triterpene, a class of compounds contained in the seeds of *S. molle* [16]. A gastroprotective effect has been demonstrated for several triterpenes [17]. The second major peak is consistent with the molecular formula of C_21_H_42_O_4_ (compound II), corresponding to an aliphatic chain with an ester functional group and hydroxyls. For such compounds, gastroprotective activity has not yet been reported.

Several possible explanations can be provided for the lower gastroprotective effect found with the white solid versus the entire F1 fraction. F1 might contain a compound capable of enhancing the activity of fractions 227–256, or there may be a sum of effects of all the compounds in the F1 fraction. It is also plausible that the aliphatic chain inhibits the biological activity of the white solid. Further research is necessary to explore these hypotheses.

Regarding the mixture of compounds in fractions 227–256, an evaluation was performed as to whether NO, prostaglandins, or sulfhydryl groups are involved in its mechanism of action. NO synthesized by NOS in endothelial cells plays an important role in maintaining the integrity of the gastric mucosa because it interacts with sensory neuropeptides and endogenous prostaglandins [18]. Since the administration of L-NAME, a non-selective inhibitor of NOS, reversed the gastroprotective effect of the white solid (Figure 4A), the NO signaling pathway is involved in its mechanism of action. In this sense, carbenoxolone (a triterpene) inhibits phosphodiesterases, which inactivate 3′,5′-cyclic monophosphate (cyclic GMP), a second messenger of the NO pathway known to promote vasorelaxation [19]. Considering that the white solid probably contains a triterpene, it likely acts in the same way as carbenoxolone by inhibiting phosphodiesterases, protecting the gastric mucosa by prolonging the activity of locally generated NO.

Prostaglandins, mainly PGE2, activate their different types of receptors to protect the gastric mucosa through mucus/bicarbonate secretion, increased blood flow, and reduced acid secretion [20]. Pretreatment with indomethacin, a COX inhibitor, reversed the effect of the white solid (Figure 4B), indicating that prostaglandins participate in its mechanism of action. Perhaps this occurs in the same manner as with carbenoxolone, which inhibits two prostaglandin-inactivating enzymes, 15-hydroxy-prostaglandin-dehydrogenase and Δ^13^-prostaglandin-reductase [19]. Hence, the white solid could possibly act in part by prolonging the activity of PGE2 to provide protection to the gastric mucosa.

On the other hand, ethanol-induced gastric damage causes a decrease in the non-protein sulfhydryl groups of the gastric mucosa because these compounds react with free radicals produced by the injured tissues. In the event that sulfhydryl groups can eliminate such free radicals, lipid peroxidation is prevented [20]. Sulfhydryl groups are also involved in maintaining the stability of disulfide bridges in gastric mucus, and consequently the integrity of the mucus layer that covers the gastric mucous membrane. The more damaged the disulfide bridges are, the more soluble the mucus will become, resulting in a greater susceptibility of the gastric mucosa to injury [21]. Pretreatment with NEM, a blocker of non-protein sulfhydryl groups, inhibited the gastroprotective effect of the white solid, indicating the probable contribution of such groups to its mechanism of action. Further research is needed to corroborate these hypotheses.

In parietal cells, the activation of the H_2_, M_3_, and cholecystokinin-2 receptors by histamine, acetylcholine, and gastrin, respectively, promotes H^+^,K^+^-ATPase activation and triggers acid secretion. The administration of the white solid did not show antisecretory activity, which indicates that it does not act as an antagonist of these receptors, nor does it block H^+^,K^+^-ATPase.

## 4. Materials and Methods

### 4.1. Animals

The experiments were performed on male Wistar rats (180–220 g) obtained from the animal house of the Universidad Nacional Autónoma de México, FES-Zaragoza, Mexico City, Mexico. The care and handling of animals was conducted in accordance with the Mexican official guidelines for laboratory animals (NOM-062-ZOO-1999) and international norms. Unless otherwise specified, the rats were placed in single cages with wire-net floors and deprived of food 24 h before experimentation. Animals were allowed free access to water throughout the experimental procedures. All assays were carried out with 7 animals per group. The study was approved by the Internal Committee for the Care and Use of Lab Animals (CICUAL, according to the initials in Spanish) of the Escuela Superior de Medicina, Instituto Politécnico Nacional, with registration number ESM.CICUAL-01/14-03-2018.

### 4.2. Drugs and Compounds

All drugs were prepared immediately before use. The hexane and dichloromethane extracts, the fractions of the hexane extract (F1, F2, and F3), and the white solid (fractions 227–256 of F1) were suspended in 0.5% Tween 80. Carbenoxolone, L-NAME, and NEM were dissolved in water, and indomethacin was dissolved in 5 mM NaHCO_3_. These compounds were purchased from Sigma Chemical Co. (St. Louis, MO, USA).

### 4.3. Plant Material

During March of 2022, the seeds of *Schinus molle* were collected in Xochitenco Chimalhuacán, in the State of Mexico, Mexico. The plant was identified and registered by Manuel de Jesús Gutiérrez Morales from the Flora Department of the Chip Herbarium, which is part of the Botanical Garden of the Secretary of Environmental Protection, Housing, and Natural History of the State of Chiapas, Mexico. A specimen of the original collection can be found with the voucher number 50327.

### 4.4. Isolation of the White Solid

The seeds of *S. molle* were dried at room temperature in the shade. After grinding 9.5 kg of the seeds, they were extracted successively by maceration at room temperature (22 ± 2 °C), first with hexane (14 L × 3) and then with dichloromethane (14 L × 3). Evaporation of the solvents under reduced pressure furnished 529 and 332 g of syrupy residue, respectively, which constituted the two extracts. Each of these extracts was given orally to rats at 30 and 100 mg/kg doses before orally administering ethanol. Given that the hexane and dichloromethane extracts showed very similar gastroprotective activity at both doses (Table 1), it was decided to continue the study with the hexane extract because of the higher yield. This extract was subjected to silica-gel column chromatography involving large differences in polarity, and therefore, fractionation. Of the resulting fractions (F1, F2, and F3), gastroprotection was detected for F1 and F2. Due to the higher yield of F1 (Table 2), it was subjected to column chromatography. The elution of fractions 227–256 with a 9:1 mixture of hexane/EtOAc (previously evaporated at reduced pressure) afforded a white solid.

Electrospray ionization (ESI) analysis was performed on the white solid in a Bruker micrOTOF-Q II mass spectrometer (Bruker Daltonics, Billerica, MA, USA). Samples were dissolved in methanol and injected directly into the spectrometer. The peaks related to the white solid were displayed in positive and negative ion mode (ESI+ or ESI−). The capillary potential was −4.5 kV, and the drying gas temperature and flow were 200 °C and 4 L/min, respectively. Ion chromatograms were obtained from *m*/*z* 500 to 3000. MS data were processed with PolyTools 1.0 (Bruker Daltonics, Billerica, MA, USA).

### 4.5. Ethanol-Induced Gastric Lesions

Gastric lesions were induced by the oral administration of absolute ethanol (1 mL), a procedure carried out 30 min after oral application of the hexane and dichloromethane extracts, the fractions of the hexane extract (F1, F2, and F3), the white solid (fractions 227–256 of F1), carbenoxolone (the reference drug), or the vehicle (0.5% Tween 80, 0.5 mL/100 g). Upon completion of another 2 h, the animals were sacrificed in a CO_2_ chamber. The stomachs were dissected and filled with 2% formalin, then placed in a container with formalin at the same concentration for 5 min. Subsequently, the stomachs were opened along the greater curvature and the gastric lesions were quantified under a stereoscopic microscope (×10) equipped with an ocular micrometer. The gastroprotection was calculated with the formula: (UIC-UIT) × 100/UIC, where UIC is the average ulcer index of the control group and UIT is the average ulcer index of the experimental group [22].

### 4.6. Ethanol-Induced Gastric Mucosal Lesions in Rats Pretreated with L-NAME

To test the possible involvement of endogenous NO in the gastroprotective activity of the mixture of compounds, L-NAME (70 mg/kg dissolved in saline) was administered intraperitoneally to 3 groups of animals. After 30 min, each group received (orally) the vehicle (Tween 80, 0.5%), the white solid (100 mg/kg), or carbenoxolone (100 mg/kg); and 30 min later, all rats were orally given ethanol, which was allowed to take effect for 2 h before sacrificing the animals to determine the ulcer index. A control group treated only with the solution of Tween 80 plus ethanol was included in this evaluation [22].

### 4.7. Ethanol-Induced Gastric Mucosal Lesions in Rats Pretreated with Indomethacin

To examine whether prostaglandins participated in the gastroprotective activity of the mixture of compounds, a control group was subcutaneously injected with NaHCO_3_ (5 mM) dissolved in saline (0.1 mL/100 g), and another group was administered indomethacin at 10 mg/kg dissolved in NaHCO_3_ (5 mM) by the same route and in the same volume. After 75 min, Tween 80 (0.5%) was given orally (0.5 mL/100 g) to the control group, and the indomethacin-pretreated group received (orally) either the mixture of compounds (100 mg/kg) or carbenoxolone (100 m/kg). Upon completion of another 30 min, all groups were orally given ethanol. The animals were sacrificed in a CO_2_ chamber 2 h later, at which time the stomachs were removed to determine the ulcer index. A group treated only with the solution of Tween 80 plus ethanol was included in this evaluation [22].

### 4.8. Ethanol-Induced Gastric Mucosal Lesions in Rats Pretreated with NEM

To explore the likely contribution of sulfhydryl groups to the mechanism of action of the mixture of compounds, NEM was injected subcutaneously to three groups (10 mg/kg). After 30 min, the rats received (orally) one of the following treatments: a Tween 80 solution, the mixture of compounds (100 mg/kg), or carbenoxolone (100 mg/kg). Upon completion of another 30 min, all animals were orally given ethanol, and 2 h later they were sacrificed to establish the ulcer index. A control group administered only Tween 80 (0.5%) plus ethanol was included in this evaluation [22].

### 4.9. Antisecretory Effect (Pylorus Ligation)

To assess secretion in the stomach, the animals were fasted for 18 h and then anesthetized with a mixture of ketamine (2 mg/mL) and xylazine (5 mg/mL) administered intraperitoneally. Immediately after, the pylorus was ligated, and the animals received (orally) one of the following treatments: Tween 80 (0.5%), the mixture of compounds (100 mg/kg), or omeprazole (30 mg/kg). They were sacrificed 4 h later to dissect the stomachs and collect the gastric content, which was centrifuged at 3000 rpm for 5 min to determine the volume and pH of the supernatant (with a pH meter).

### 4.10. Statistics

Data are expressed as mean ± SEM (*n* = 7). The differences between treatment groups were examined for statistical significance by the Kruskal–Wallis test, followed by Dunn’s multiple comparison, with significance considered at *p* ≤ 0.05. The Mann–Whitney *U* test was employed to compare two groups.

## 5. Conclusions

Scientific evidence is herein presented for the first time of the gastroprotective activity of *S. molle*, a plant long used to treat gastric disorders in Mexican traditional medicine. The dichloromethane and hexane extracts each showed a nearly 100% gastroprotective effect in the rat model of ethanol-induced gastric lesions. From the hexane extract, fractions 227–256 provided a white solid which is a mixture of compounds and an active gastroprotective agent. Of the two principal compounds in the mixture, the predominant one is probably a triterpene. Gastroprotective activity has been demonstrated for several triterpenes. The other major compound is an aliphatic chain with ester and hydroxyl functional groups, for which a gastroprotective effect has not been reported. NO, prostaglandins, and non-protein sulfhydryl groups are involved in the mechanism of action of this mixture of compounds. No antisecretory effect was detected.

## Figures and Tables

**Figure 1 molecules-27-07321-f001:**
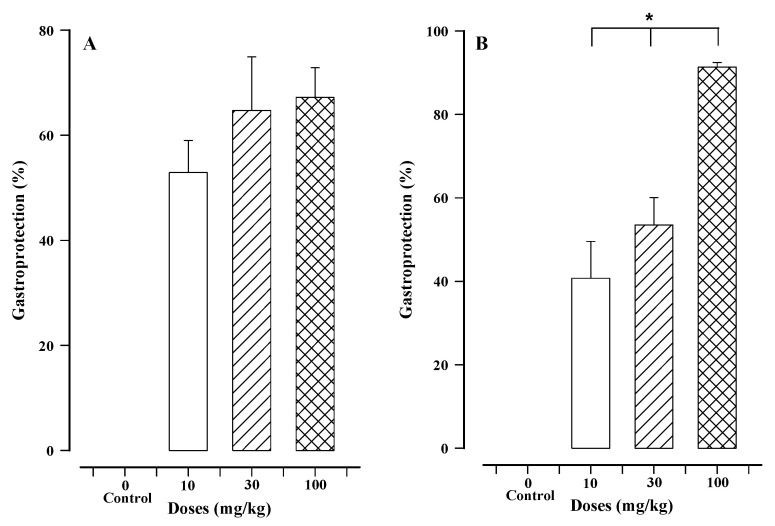
The gastroprotective effects of the white solid (**A**) and carbenoxolone (**B**). Bars represent the mean ± SEM (*n* = 7). * *p* < 0.05, utilizing the Kruskal–Wallis test followed by Dunn’s multiple comparison.

**Figure 2 molecules-27-07321-f002:**
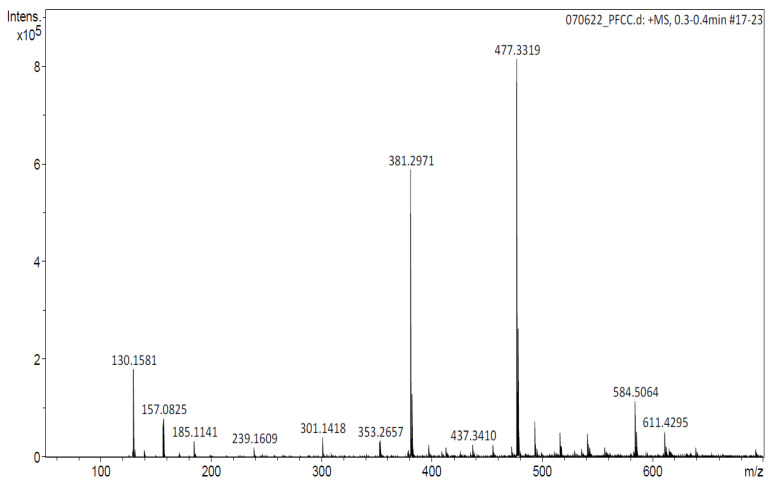
Electrospray spectrum (in the positive mode) for the white solid (fractions 227–256 of F1).

**Figure 3 molecules-27-07321-f003:**
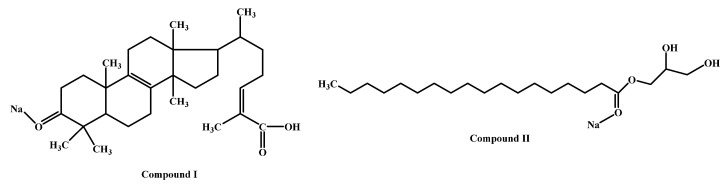
The structures of the proposed compounds based on the data from the ESI-MS analysis.

**Figure 4 molecules-27-07321-f004:**
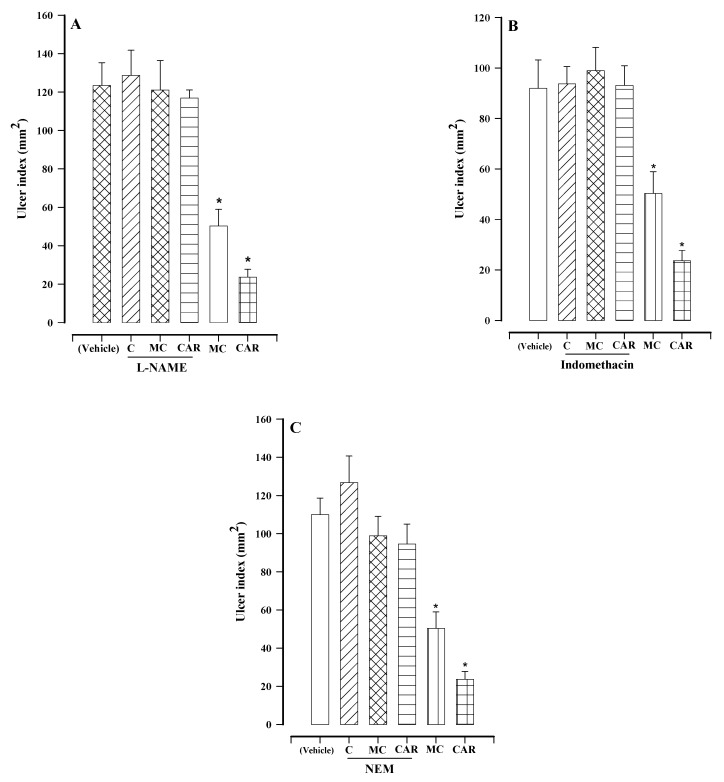
Regarding ethanol-induced gastric lesions in rats, the effects of two treatments are shown, the white solid (fractions 227–256 of F1) and carbenoxolone, each administered after one of three pretreatments: (**A**) L-NAME (70 mg/kg), (**B**) indomethacin (10 mg/kg), or (**C**) NEM (10 mg/kg). Bars represent the mean ± SEM (*n* = 7). * *p* < 0.05 vs. the respective control, based on the Kruskal–Wallis test, followed by Dunn’s multiple comparison. C = control inhibitors, MC = the mixture of compounds (the white solid), and CAR = carbenoxolone.

**Table 1 molecules-27-07321-t001:** Gastroprotective effects of the hexane and dichloromethane extracts.

Treatment	Dose (mg/kg)	*n*	Gastroprotection (%)
Hexane extract	30	7	75.57 ± 3.65 *
	100	7	99.03 ± 0.28
Dichloromethane extract	30	7	76.91 ± 1.44 ^#^
	100	7	99.32 ± 0.28
Carbenoxolone	100	7	94.90 ± 0.41

Data represent the mean ± SEM (*n* = 7). Mann–Whitney U test. * *p* < 0.05 compared with 100 mg/kg of hexane extract, ^#^
*p* < 0.05 compared with 100 mg/kg of dichloromethane extract.

**Table 2 molecules-27-07321-t002:** Gastroprotective effects of the fractions of the hexane extract on ethanol-induced lesions.

Treatment	Dose (mg/kg)	*n*	Gastroprotection (%)
F1	100	7	99.37 ± 0.44
F2	100	7	99.62 ± 0.25
F3	100	7	37.24 ± 4.13 *
Carbenoxolone	100	7	99.67 ± 0.21

Data represent the mean ± SEM (*n* = 7). Dunn’s multiple comparison test after the Kruskal–Wallis test. * *p* < 0.05 compared with F1, F2, and carbenoxolone.

**Table 3 molecules-27-07321-t003:** Evaluation of the antisecretory activity of the white solid.

Treatment	Dose (mg/kg)	*n*	Volume (mL)	pH
Control	0	7	1.66 ± 0.04	2.19 ± 0.06
White solid	100	7	1.27 ± 0.28	2.09 ± 0.08
Omeprazole	30	7	1.57 ± 0.10	5.98 ± 0.24 *

* *p* < 0.05, utilizing the Kruskal–Wallis test, followed by Dunn’s multiple comparison.

## Data Availability

Not applicable.

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
