# Peer review of "First Evidence of Gastroprotection by Schinus molle: Roles of Nitric Oxide, Prostaglandins, and Sulfhydryls Groups in Its Mechanism of Action"

_molecules, 2022, doi:10.3390/molecules27217321_

Round 1

Reviewer 1 Report

The authors of the study ‘First evidence of gastroprotection by Schinus molle: Role of nitric oxide, prostaglandins, and sulfhydryls groups in its mechanism of action assessed gastroprotective effect of plant extracts from Schinus molle on ulcer in Wistar rats. They observed that main compounds present in extracts, such as triterpene and a compound with aliphatic chain with ester and hydroxyl functional groups had the predominant gastroprotective effect. The authors also suggested possible involvement of nitric oxide, prostaglandins, and sulfhydryl groups in the gastroprotective protection of the tested plant extracts.

Generally, the study is quite interesting, the methods have been properly selected
and the results are convincing. Nevertheless, the authors must address some comments to improve the manuscript.

Comments

The authors should explain the reason of using the values of plant extract doses in their experiments

Results, Figure 1, it would be much reasonable to compare statistically each doses of white solid (Fig. 1A) with doses of a reference drug (Fig 1B). Figure 1A and 1B should be combined to shown this comparison.

Abstract, line 5-6, correct to: The hexane and dichloromethane extracts from tested plant …

Results, line 11, what does it mean that fraction F3 was inactive, please explain.

Results, line 1-2, correct to: Oral administration of hexane and dichloromethane extractsof S. molle seeds at doses of 30 mg/kg and 100 mg/kg inhibited …

Discussion, line 54, correct to: … these compounds react with free radicals produced by the injured tissues.

4.4. Isolation of white solid, line 12, Was hexane/EtOAc mixture evaporated to achieve a white solid?, please explain.

4.5. Ethanol-induced lesions, line 4, specify what was a vehicle?

Author Response

The authors of the study ‘First evidence of gastroprotection by Schinus molle: Role of nitric oxide, prostaglandins, and sulfhydryls groups in its mechanism of action’ assessed gastroprotective effect of plant extracts from Schinus molle on ulcer in Wistar rats. They observed that main compounds present in extracts, such as triterpene and a compound with aliphatic chain with ester and hydroxyl functional groups had the predominant gastroprotective effect. The authors also suggested possible involvement of nitric oxide, prostaglandins, and sulfhydryl groups in the gastroprotective protection of the tested plant extracts.

Generally, the study is quite interesting, the methods have been properly selected and the results are convincing. Nevertheless, the authors must address some comments to improve the manuscript.

Comments

1.- The authors should explain the reason of using the values of plant extract doses in their experiments

Response: In the present study, the dose of 100 mg/kg of the extracts was tested in order to be able to compare the results to those reported in previous studies with carbenoxolone (the reference compound), a dose at which the latter reached 100% of gastroprotection when using the same animal model. The dose of 30 mg/kg was evaluated to determine whether the extracts followed the same behavior as carbenoxolone in regard to the 50% protective effect.    

2.- Results, Figure 1, it would be much reasonable to compare statistically each doses of white solid (Fig. 1A) with doses of a reference drug (Fig 1B). Figure 1A and 1B should be combined to shown this comparison.

Response: We thank the reviewer for the suggestion. The comparison between the white solid and carbenoxolone was made at each dose, finding a statistically significant difference only at the dose of 100 mg/kg, and not at 30 or 10 mg/kg. These comparisons are now included in the text in the paragraph in section 2.3, “Gastric protective activity of the white solid” (p. 4). On the other hand, we consider that showing the graphs separately is more convenient to clearly illustrate the lack of a dose-dependent effect of the white solid, a behavior distinct from that found with carbenoxolone.

3.- Abstract, line 5-6, correct to: The hexane and dichloromethane extracts from tested plant …

Response: The suggested change has been made.

4.- Results, line 11, what does it mean that fraction F3 was inactive, please explain

Response: The following sentence was added to the text in section 2.2 to clarify this point: “F3 was considered inactive, having provided less than 50% gastroprotection (Table 2).”

5.- Results, line 1-2, correct to: Oral administration of hexane and dichloromethane extractsof S. molle seeds at doses of 30 mg/kg and 100 mg/kg inhibited …

Response: The suggested change was made.

6.- Discussion, line 54, correct to: … these compounds react with free radicals produced by the injured tissues.

Response: The suggested change was made.

7.- 4.4. Isolation of white solid, line 12, Was hexane/EtOAc mixture evaporated to achieve a white solid?, please explain.

Response: Indeed, the hexane/EtOAc mixture was evaporated in a rotary evaporator to obtain the solid. To clarify this point, a phrase was added to the following sentence (p. 10): “The elution of the fractions 227-256 with a 9:1 mixture of hexane/EtOAc (previously evaporated at reduced pressure) afforded a white solid.

8- 4.5. Ethanol-induced lesions, line 4, specify what was a vehicle?

Response: The suggested change was made and can be found on p. 10: “the vehicle (0.5% Tween 80, 0.5 mL/100 g)”.

Reviewer 2 Report

Schinus molle is quite an interesting drug, but the authors didn't provide proper data to address any mechanism. No control, no criteria define the EtOH-induced ulceration.  No IHC staining to really show the protection by Schinus molle. L-NAME, Indomethacin, and NEM still show a correlation between ulceration, no special knowdown or inhibitor of COX2 or iNOS to evaluate the effect of EtOH-induced ulceration,  meanwhile which cells were affected in this protection progress?

Author Response

Schinus molle is quite an interesting drug, but the authors didn't provide proper data to address any mechanism.

Response: We beg to differ with the reviewer on this point. The cytoprotective effect and antisecretory effect were evaluated. Regarding the cytoprotective effect, three factors were tested: nitric oxide, prostaglandins, and sulfhydryl groups, finding all three implied in the gastroprotective effect of the test compound. Hence, gastroprotection cannot be assigned to a single factor, as has been found for carbenoxolone as well. The present results provide the foundation for continuing to study the plant extracts using specific inhibitors. On the other hand, an antisecretory effect was ruled out based on the results of the assay based on the model of a pylorus ligation.

No control, no criteria define the EtOH-induced ulceration.  No IHC staining to really show the protection by Schinus molle.

Response: Control groups (using the vehicle or the reference compound) were included in all evaluations. In section 4.5 (Ethanol-induced gastric lesions) of the Methods section, the experimental model is explained and it is mentioned that a control group was required to calculate the percentage of gastroprotection represented in the graphs. Moreover, carbenoxolone was employed as the reference compound in all evaluations that implied the cytoprotection model. In the antisecretory assay (section 4.9, “Antisecretory effect”), the reference drug was omeprazole and a vehicle control group was included (Table 3).

The model of ethanol-induced gastric lesions is widely used to study the efficacy of potential drugs or assess cytoprotective and/or antioxidant agents. Please see: M. B. Adinortey, C. Ansah, I. Galyuon, and A. Nyarko. In Vivo models used for evaluation of potential antigastroduodenal ulcer agents, Ulcers, vol. 2013, Article ID796405, 12 pages, 2013. Indeed, the studies carried out with IHC staining are of great help in this type of investigation. However, this technique was not considered necessary presently due to the testing herein of a mixture of compounds. In a future study, when the compound responsible for the activity is obtained, IHC staining will be considered.

L-NAME, Indomethacin, and NEM still show a correlation between ulceration, no special knowdown or inhibitor of COX2 or iNOS to evaluate the effect of EtOH-induced ulceration, meanwhile which cells were affected in this protection progress?

Response: This study provides the first evidence of the gastroprotection of Shinus mole. Unspecific inhibitors of NO, prostaglandins, and sulfhydryl groups were used to explore possible mechanisms of action. Given that the results of the present study have established that these three factors are involved in the gastroprotective effect, the foundation of evidence has been established to continue this line of research in the future by utilizing specific inhibitors that will allow us to further analyzed the action mechanism. It is well known that ethanol-induced gastric lesions damage the cells of the gastric mucosa. Please see: Qin, Shumin; Huang, Keer; Fang, Zhigang; Yin, Jinjin; Dai, Ruwei (2017). The effect of Astragaloside IV on ethanol-induced gastric mucosal injury in rats: Involvement of inflammation. International Immunopharmacology, 52, 211–217. Hence, the study of this was not a priority in the present work.

Round 2

Reviewer 2 Report

1,In Figure1, please include the control group.

2, As the authors claimed, they tested nitric oxide, prostaglandins, and sulfhydryl groups, to reflect cytoprotective effects, please list them in the table.

3, After EtOH treatment, what is the behavior of the rats? and the ages of the test rats?

4, How about the blood test with white solid treatment vs control groups?

Author Response

1.- In Figure1, please include the control group.

Response: The suggested change has been made.

2.- As the authors claimed, they tested nitric oxide, prostaglandins, and sulfhydryl groups, to reflect cytoprotective effects, please list them in the table.

Response: In the evaluations carried out to determine the participation of: nitric oxide, prostaglandins and sulfhydryl groups in the mechanism of action of the white solid. L-NAME (non-specific inhibitor of nitric oxide synthases), indomethacin (non-specific inhibitor of COX) and NEM (a blocker of sulfhydryl groups) were used. This is mentioned in the results discussion section, paragraphs 6-8. The results are shown in Figure 4, so representing them in a table would be repetitive. It is important to point out that in the present study the concentration of these protective factors was not determined.

3, After EtOH treatment, what is the behavior of the rats? and the ages of the test rats?

Response: The behavior of the rats after the administration of ethanol is that attributed to the ingestion of a central nervous system depressant. The age of the animals is 8 weeks

Response: 4.- How about the blood test with white solid treatment vs control groups?

The present study only includes the pharmacodynamic aspect and not the pharmacokinetic aspect in which the quantitative determination in biological fluids or tissues (eg blood) of the compound of interest is crucial.
